# S-WD-EEMD: A hybrid framework for imbalanced sEMG signal analysis in diagnosis of human knee abnormality

Ankit Vijayvargiya[1,2]*, Aparna Sinha[3], Naveen Gehlot[4], Ashutosh Jena[4], Rajesh Kumar[4], Kieran Moran[1]

1 Insight Science Foundation Ireland Research Centre for Data Analytics, School of Human and Health Performance, Dublin City University, Dublin, Ireland, 2 Department of Electrical Engineering, Swami Keshvanand Institute of Technology, Management & Gramothan, Jaipur, Rajasthan, India, 3 Department of Information Technology, Bansthali Vidyapeeth, Radha Kishnpura, Rajasthan, India, 4 Department of Electrical Engineering, Malaviya National Institute of Technology, Jaipur, Rajasthan, India

* ankit.vijayvargiya@dcu.ie

**Data Availability Statement:** The data is publicly available at doi: 10.24432/C5ZW3P and doi: 10. 6084/m9.figshare.25391458.

## Abstract

The diagnosis of human knee abnormalities using the surface electromyography (sEMG) signal obtained from lower limb muscles with machine learning is a major problem due to the noisy nature of the sEMG signal and the imbalance in data corresponding to healthy and knee abnormal subjects. To address this challenge, a combination of wavelet decomposition (WD) with ensemble empirical mode decomposition (EEMD) and the Synthetic Minority Oversampling Technique (S-WD-EEMD) is proposed. In this study, a hybrid WD-EEMD is considered for the minimization of noises produced in the sEMG signal during the collection, while the Synthetic Minority Oversampling Technique (SMOTE) is considered to balance the data by increasing the minority class samples during the training of machine learning techniques. The findings indicate that the hybrid WD-EEMD with SMOTE oversampling technique enhances the efficacy of the examined classifiers when employed on the imbalanced sEMG data. The F-Score of the Extra Tree Classifier, when utilizing WD-EEMD signal processing with SMOTE oversampling, is 98.4%, whereas, without the SMOTE oversampling technique, it is 95.1%.

## 1 Introduction

Knee pain is a common problem affecting numerous individuals, significantly impairing their ability to participate in various daily activities. The knee joint, a synovial joint in our body, comprises three bones: the femur, tibia, and patella. It is fortified by ligaments, tendons, muscles, and cartilage. The knee joint not only provides stability for a wide range of motion but also supports various activities (e.g., walking, running, jumping, and bending) and acts as a shock absorber. As supported by a survey [1], one out of every four individuals aged 18–24 experiences knee problems resulting from various causes, including knee sprains, fractures, and ligament tears, and conditions, such as tendonitis, arthritis, cerebral palsy, or degenerative

**Funding:** Ankit Vijayvargiya received funding for this research article from the EuropeanUnion's Horizon and Innovation Programme under the NeuroInsight Marie Sklodowska-Curie under grant number 101034252. The funders had no role in study, design, data collection and analysis, decision to publish or preparation of the manuscript.

**Competing interests:** The authors have declared that no competing interests exist.

joint disease. A timely and accurate diagnosis of these abnormalities is crucial for effective treatment planning and rehabilitation.

Knee disorders have been diagnosed in the past using a combination of clinical evaluation, patient history, physical examination, and imaging modalities. During clinical evaluation, the patient's stated symptoms, such as pain, swelling, and reduced range of motion, are considered. During a physical check, inspection, palpation, and specialized orthopedic tests are used to look for particular signs of knee problems [2]. For the clinical identification of these diseases, imaging methods of Magnetic Resonance Imaging (MRI), X-rays, Computed Tomography (CT) scans, and angiography are also used [3, 4]. Even though these old ways of diagnosing have been useful, they have their challenges. They may be subjective, depend on the skill and experience of the doctor, and take more time and resources for imaging visits. Also, some knee problems may not have clear physical signs or differences on the first images, which means that more research or follow-up exams are needed. Advances in medical technology and the development of new testing methods, such as bio-mechanical analysis and computational modeling, have made it easier to find knee problems early and accurately [5].

The incorporation of artificial intelligence (AI) in the medical sector enhances the quality and speed of medical consultation [6]. When it comes to detecting knee issues, AI proves to be helpful in various aspects, including improved accuracy, time efficiency, assisting healthcare professionals, standardization of diagnosis, early detection and prevention, and augmenting limited resources. AI algorithms analyze large datasets with precision, consistency, and faster speed than humans, thereby improving the accuracy of detecting knee issues [7]. AI systems can pick up on minor differences or abnormalities that humans might miss or not notice. This makes it less likely that a diagnosis will be delayed or incorrect [8]. In time-sensitive situations, the speed of detection is especially important because it allows for quick recognition and analysis. AI can conduct automated analyses, assisting radiologists and orthopedic experts in making diagnoses [9]. This could make their jobs easier and help them work more efficiently overall. There is a lack of radiologists and orthopedic experts in many healthcare systems. AI can help a lot with this problem by adding another level of analysis and screening for knee problems, especially in places where it's hard to get specialized care. AI can help a lot when it comes to finding knee problems, but it can't replace the judgment and experience of doctors and nurses. AI is used by healthcare workers so that they can improve their skills, work more efficiently, and give better care to their patients.

Electromyography (EMG) is employed by healthcare practitioners to quantitatively assess and record the electrical signals generated by skeletal muscles. EMG signals give important information about how muscles work, their amplitude and timing of activation, and if there are any problems or abnormalities. They also play a crucial role in diagnosis, rehabilitation, treatment planning, and research. There are two methods often employed for the acquisition of EMG data, namely surface EMG (sEMG) and intracellular EMG (iEMG). sEMG includes placing electrodes on the skin right above the muscle of interest. These electrodes pick up and record the electrical energy that is produced by muscles under the skin. sEMG is a useful and non-invasive way to measure muscle function and activity, while iEMG involves inserting needle probes directly into the muscle tissue to record electrical activity. Both sEMG and iEMG are useful in their ways. sEMG is less invasive, which makes it better for regular checks and studies on larger scales. Most of the time, sEMG is better than iEMG for finding issues with the knee. It is preferred because it is easy to use, less risky for the patient, and can measure a wide range of muscle action patterns [10]. sEMG is a useful tool for clinical screening and therapy because it tells us a lot about how knee problems affect muscle activation, coordination, and abnormalities [11].

sEMG signals have been used to make a wide range of applications, such as exoskeletons or muscle signal actuated systems for upper and lower limb prosthetics [12], as well as to find neuromuscular diseases [13] and monitoring of fitness and exercise performance [14]. The outcomes of machine learning algorithms are shown to correlate with six different lower-limb movements that are classified by Khimraj et al. [15]. Based on the time-domain (TD) features of the signals, Hudgins et al. classified four distinct types of limb movements that are measured by sEMGs using multilayer perceptron neural networks [16]. Huang et al. [17] used a back-propagation neural network (BPNN) and autoencoders to estimate the human arm joint force from sEMG data. Silva et al. [18] investigated a spinal cord injury by using sEMG signals that are obtained during upper limb movement. Sudarsan et al. used sEMG signals to control an artificial limb that was designed and developed [19]. Hand motions may be identified from sEMG signals in amputees, as indicated by Tuncer et al. [20] by the extraction of discrete wavelet features utilizing ternary patterns. Cai et al. [21] explored five subjects utilizing Support Vector Machine (SVM) based classification for diverse upper appendage developments. Chandra Prakash et al. [22] proposed a linear time-series-based prediction model that could be used to control robotic assistive devices to increase the strength of the lower extremities.

Researchers have devoted the last decade focusing on exploiting sEMG signals to identify lower limb activities or detect lower limb disorders [23–25]. Chen et al. used deep belief networks and sEMG signals to estimate flexion and extension joint angles of the human lower limb [26]. Bonato et al. studied the effect of exhaustion of hamstring muscles and quadriceps on sEMG activity [27]. According to Swaroop et al., myopathy and neuropathy can be classified through sEMG signals using a three-layered neural network with backpropagation [28]. In a study by Kugler et al. [29], EMG data was used for the identification of rare Parkinson's disease via SVM-based methods. Morbidoni et al. utilized a deep learning methodology to classify the gait phases based on sEMG data collected during walking [30].

Analyzing sEMG based signals collected from the muscles is difficult due to the presence of noisy signals that include ambient and inherent noise from the sensing device, crosstalk and motion artifacts from the irregular muscular movements [31]. In such cases, denoising techniques that involve wavelet decomposition [23], empirical mode decomposition [32], or independent component analysis [33] can prove to be essential for the minimization of noise. In previous studies [34, 35], the authors introduced a mix of denoising methodologies known as WD-EEMD (Wavelet Decomposition with Ensemble Empirical Mode Decomposition). It assists in the examination of sEMG signals during the identification of lower limb and upper limb activities. This hybrid approach demonstrated superior performance compared to individual decomposition strategies. The method of wavelet decomposition (WD) is employed in signal processing to partition a signal into discrete sub-bands of varying frequencies. The technique allows for the examination of the signal at several resolutions, facilitating investigation at different scales. The process of wavelet decomposition necessitates the utilization of wavelet analysis, which utilizes wavelet functions to examine the time-frequency characteristics of a given signal. Various wavelet scales are used to distinguish between different frequency bands, and the wavelet transform effectively partitions the signal into discrete frequency segments [36]. EMD is a widely employed signal processing approach that facilitates the decomposition of signals into their constituent intrinsic mode functions (IMFs). During the latter part of the 1990s, Huang et al. [37] introduced a data-driven methodology for the analysis of signals that exhibit nonlinearity and nonstationarity. This processing technique involves the decomposition of a raw signal into finite representative components called the Intrinsic Mode Functions (IMFs). These IMFs are subsequently averaged to obtain the final reconstructed signal. This is called the EMD technique and the many intrinsic mode functions (IMFs) as a result of decomposition can effectively extract the diverse oscillating components of the signal occurring at

different scales. The individual IMF components are organized based on their respective scales, with the greatest frequency oscillations being presented first and the lowest frequency components being presented last. A wavelet denoiser can effectively eliminate white gaussian noise (WGN) from the desired range of sEMG signal frequency while preserving its key features. At the same time, it is also found to be very effective against the interfering noise from nearby muscles. Ensemble Empirical Mode Decomposition (EEMD) [38], on the other hand, can filter out the interference from power lines (PLI) and the disturbances due to baseline wandering (BW). WD-EEMD, which combines wavelet decomposition and EEMD, is an effective method for minimizing various noises in the sEMG signal.

Class imbalance is a prevalent occurrence in numerous real-world scenarios, such as anomaly detection, fraud detection, medical diagnosis, and rare event prediction [39, 40]. It arises when the categories or classes of the target variable in classification problems are not evenly represented in the dataset. In simpler terms, one class or category may have a significantly larger number of instances compared to the other, or vice versa. Addressing the class imbalance problem is crucial, as it presents challenges for predictive modeling and can impact the performance and accuracy of the models. It can particularly affect the performance of the minority class, as the model tends to be biased towards the majority class [41]. Akbar et al. proposed a computational model, iAFP-gap-SMOTE in distinguishing antifreeze proteins (AFPs) from non-AFPs [42].

The imbalance issue in the obtained sEMG data may occur during the act of walking, mostly owing to variances in data length observed between people with abnormal knee problems and those with healthy knees [43]. Individuals who have knee abnormalities tend to exhibit prolonged completion times for movement tasks, leading to an extended sEMG signal duration in comparison to that of individuals without such abnormalities. Hence, the considerable disparity in signal duration observed between individuals with abnormal conditions and those who are healthy presents a challenge of class imbalance within the sEMG data acquired during the activity of walking. Dataset resampling is the prevailing method utilized to address this issue. This process can encompass either oversampling the minority class by the generation of synthetic samples or duplication of existing instances or undersampling the majority class by eliminating instances from the dataset, to attain a balanced distribution of classes [44]. In our previous study [43], the Synthetic Minority Over-sampling Technique (SMOTE) is used for balancing the imbalanced sEMG dataset by synthesizing data corresponding to lower limb signals from healthy and knee abnormalities. Furthermore, it is shown that the identification process is improved. It works on the principle of resampling. It generates artificial samples for the minority class by interpolating between existing minority class samples. SMOTE helps in balancing the class distribution by creating synthetic instances that are similar to the minority class more accurately. The results obtained during that study suggest that SMOTE oversampling techniques enhance the performance of classifiers using sEMG data with imbalanced distributions.

In this study, the authors have proposed a hybrid framework that combines WD-EEMD signal pre-processing with the SMOTE oversampling technique to detect individuals with knee injuries using imbalanced sEMG data. The important contributions made in this study are listed as follows:

- Develop a hybrid framework (S-WD-EEMD) for identifying knee disorders using imbalanced sEMG data while accounting for variations in signal time frames between normal and abnormal individuals.

- A hybrid of WD and EEMD is used to minimize the noise present in the sEMG signals collected from the lower limb muscles.

- A window followed by an overlap is traversed over the signal from which eleven time-domain (TD) features of sEMG signals are obtained.

- A SMOTE oversampling technique is used to enhance the extracted features of the minority class data (healthy individuals).

- Performance parameters of machine learning classifiers are calculated for the identification of knee abnormal individuals with and without oversampled data.

The article is organized as follows: The problem along with the literature overview has been introduced in Section 1. Section 2 is about the dataset considered during the study, and section 3 discusses the materials and the proposed methodology. The results and discussions of the proposed models are included in section 4. The study is concluded along with some recommendations for the future in section 5.

## 2 Dataset

This study used sEMG signal data publicly available in the UCI-hosted machine learning repository by Sanchez et al. [45]. This dataset contains the sEMG signals of 22 subjects over the age of 18, 11 of whom had injuries or pain to their knees, and 11 of whom were healthy. Anterior cruciate ligament injuries were reported in six abnormal subjects, meniscus injuries in four, and sciatic nerve injuries in one. However, there is no record of previous injury in the healthy subjects. As the subjects flexed their legs up, extended their legs from a sitting position, and walked, data was recorded with the help of a goniometer and DataLog MWX8 datalogger (Biometrics Ltd.). The recorded sEMG data was collected from lower limb muscles, i.e., vastus medialis (VM), semitendinosus (ST), rectus femoris (RF), and biceps femoris (BF), using a goniometer connected near the outer part of the joint. Those with abnormal knees and healthy subjects with the left leg were selected for acquiring EMG signals. The data acquired had a 14-bit resolution and was sampled at a rate of 1000 samples per second. sEMG signals were filtered through a bandpass filter allowing only 20 Hz to 460 Hz. A transition state data set was not collected, e.g., sitting to walking, standing to sitting, walking to standing, etc. The MWX8 device was used to transfer the data directly to the computer using Bluetooth. The present study focused on the issue of imbalances in sEMG data during walking, so it used the sEMG signals acquired only while the healthy and abnormal individuals walked and not the transition data.

## 3 Proposed methodology

In this section, an overview of the procedure corresponding to the execution of the study for detecting knee abnormalities using imbalanced sEMG signal data is discussed acquired during walking. Fig 1 depicts the overall procedure describing the execution of the proposed methodology.

### 3.1 Wavelet Denoising (WD)

A significant characteristic of wavelet denoising is that it preserves the important features of the signal while simultaneously removing noise from it through wavelet analysis. Signals with both low-frequency components of interest and high-frequency noise can be easily processed using this method. Wavelet denoising comprises signal decomposition, thresholding, and reconstruction. First, the signal is decomposed using wavelet analysis into frequency subbands and then combined to produce a signal. By fitting and compressing the signal iteratively, different levels of approximation and detail coefficients can be obtained by multilevel

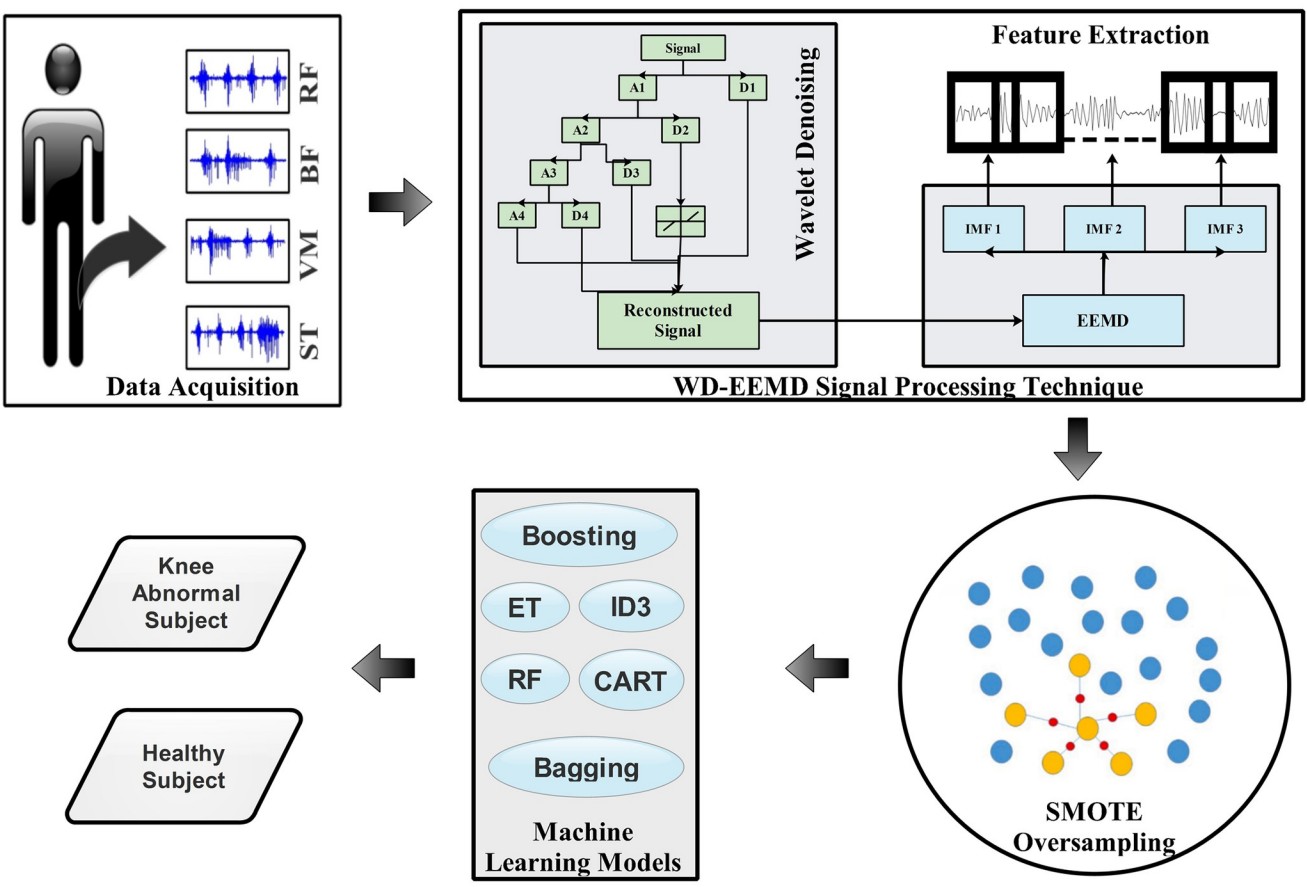

**Fig 1. Proposed approach for detecting abnormalities in knees from sEMG signals recorded during walking.**

wavelet decomposition [36]. A thresholding technique is applied to wavelet coefficients after the signal has been decomposed. As a result of this method, the coefficients representing the important aspects of a signal are preserved, while the coefficients representing noise are removed. A variety of thresholding methods are available, including soft thresholding, hard thresholding, and adaptive thresholding [46]. As a result of soft thresholding, wavelet coefficients below a threshold are zeroed, and the rest of the coefficients are reduced by a specified factor, while with hard thresholding, wavelet coefficients below a set threshold are equated to zero. By adapting the threshold value according to the wavelet coefficients or the signal's statistical properties, adaptive thresholding combines the characteristics of hard and soft thresholding [47]. The white Gaussian noise Z(n) is usually represented as:

$$Z(n) = X(n) + K(n) \tag{1}$$

where the original signal is represented by X(n) and the noise is represented by K(n). Reconstruction involves regenerating the denoised signal using the modified wavelet coefficients after thresholding using the inverse wavelet transform. As a result of combining the approximation coefficients and modified details, the denoised signal is obtained. In addition to handling both high-frequency and low-frequency noise effectively, this method has several other advantages, such as maintaining signal features at different scales. The non-parametric method makes no assumptions about the distribution of noise, making it the superior

method [48]. The universal threshold utilized during the study can be defined using the following equation:

$$\lambda = \sigma\sqrt{2lnN} \tag{2}$$

here, $\sigma$ defines the ratio of Mean Absolute Deviation to numerical value 0.6745, and N defines the signal length.

## 3.2 Empirical Mode Decomposition

Empirical Mode Decomposition (EMD) is a method for the analysis of signals that involves the computation of IMFs. In doing so, the complexity of the signal is divided into elementary oscillatory components. This technique was introduced in 1998 by Huang et al. [37] and is particularly useful for analyzing nonlinear and non-stationary signals. One of the greatest advantages of EMD lies in its ability to decompose the signals in the time domain, which cannot be done with standard Fourier or wavelet-based approaches. It employs a data-driven methodology that allows it to analyze signals with variable frequencies and amplitudes. EMD offers a data-adaptive and localized decomposition that allows for the examination of signal components based on their scales. Various ways of analyzing the IMFs acquired through EMD can be used to understand the underlying dynamics of a signal. The original signal is obtained by summation of IMFs:

$$y(t) = \sum_{n=1}^{N-1} IMF_n(t) + a_N(t) \tag{3}$$

where $a_N$ represents the residual after the extraction of $N-1$ IMFs.

While EMD has grown in popularity, it does have certain drawbacks, one of which is mode mixing, which refers to the depiction of comparable oscillatory patterns in multiple IMFs [49]. End-effects, which add artifacts at the signal border, are another drawback [50]. Researchers have suggested different modifications and alternatives, such as EEMD and Complementary EEMD with Adaptive Noise (CEEMDAN), to increase accuracy and overcome these constraints [51]. This work considers EEMD, which solves the problems associated with the mixing of modes and end effects. A noise-assisted approach is used in EEMD to deconstruct a signal into IMFs. This is accomplished by repeatedly adding white noise to the original signal and then decomposing the resultant noisy signal with EMD.

$$y_n(t) = y(t) + r_n(t) \tag{4}$$

here $r_n(t)$ is the $n^{th}$ instance of white noise and $y(t)$ is the original signal. $y_n(t)$ is the resultant noisy signal made out of the combination of noise on top of the original signal. The decomposition of different noisy signals is then averaged to get the final IMF components.

$$IMF_q = \frac{1}{T}\sum_{1}^{T} h_{q,n}, \qquad q = 1, 2, ..., Q \qquad n = 1, 2, ..., T \tag{5}$$

By calculating the average of the noisy signals, EEMD helps in the reduction of the mode mixing and end effects. The EEMD approach improves the credibility of the decomposition results and robustness; it also provides a better representation of the underlying signal modes.

### 3.3 WD-EEMD

WD-EEMD is a hybrid technique for signal processing that incorporates wavelet denoising and EEMD. EEMD is utilized to eradicate low-frequency noise, whereas wavelet denoising is

utilized to remove high-frequency noise. Particularly for non-stationary signals, the combined implementation of these techniques can be more effective than using them individually. The process begins with the application of wavelet denoising to reduce signal disruption. Then, EEMD decomposition is performed on the signal that has been denoised. By minimizing noise before decomposition, WD-EEMD seeks to more precisely reveal the signal's underlying components. Through a wavelet transform, the signal is decomposed into wavelet coefficients in wavelet analysis. Nevertheless, these coefficients frequently include high-frequency disruption. In response, the high-frequency wavelet coefficients are thresholded or eliminated, effectively removing the high-frequency noise and improving signal quality. Using EMD, the remaining wavelet coefficients are decomposed into a succession of IMFs. These IMFs are subsequently filtered or thresholded to eliminate any remaining noise in the low frequency of the signal, which is essential to eliminate noise and enhance the signal. The hybrid approach of wavelet denoising and EEMD has been effectively applied to denoise an EMG signal for a variety of activities, including lower limb activities, and upper limb activities [34, 35]. In certain circumstances, this technique can be more effective than wavelet denoising or EEMD alone, despite requiring additional computational resources [34].

In this study, several subsets of sEMG signals are obtained through wavelet decomposition in the time-frequency domain (TFD), each of which contains a series of coefficients representing the time domain evolution of the signal along with its associated frequency range. This method is implemented by setting the mother wavelet parameter to db7 from the Daubechies family at the fourth level to decompose the raw sEMG signal [52]. At the end of decomposition, a single approximate coefficient is obtained from the last level and four detailed coefficients are obtained from all four levels of decomposition to reconstruct the signal. Garotte thresholding is applied with a universal selection rule on the second detail coefficient level (D2) [35]. After that, a reconstructed signal from WD is processed with EEMD which decomposes the signal further into a set of IMFs with a residual. Each IMF consists of either a single frequency or bands of limited frequency that allow a better frequency domain (FD) representation of the sEMG signal. Finally, three IMFs (IMF1, IMF2, IMF3) are considered for the feature extraction.

### 3.4 Feature extraction

The conventional approach to applying a machine learning model relies on handcrafted features [53–55]. This method has garnered widespread acceptance within the research community and has yielded remarkable results on numerous widely recognized public datasets. It involves extracting meaningful features from a signal and subsequently employing a generic machine learning classifier, such as a Support Vector Machine (SVM) or Linear Discriminant Analysis (LDA), for classification purposes.

In this study, following the preprocessing stage involving WD-EEMD based noise reduction, a window of length 256ms is traversed along the signal. Features are extracted over this window to avoid misclassification due to the stochastic nature of the signal. This technique is referred to as a sliding window technique, which can be either adjacent or overlapping [31]. To improve the accuracy and reduce the computational latency of a classifier, nine time domain features are extracted from the signal. These features are used as inputs to the classifier model. An assessment of knee abnormalities using sEMG signals recorded during walking has been carried out using eleven time-domain features [56].

**3.4.1 Mean Absolute Value (MAV).** Taking the absolute values of the signal ($y_j$) and averaging of $M$ value together, we get the mean absolute value. Signal variability is measured by

this parameter.

$$MAV = \frac{1}{M} \sum_{j=1}^{M} | y_j |$$  (6)

### 3.4.2 Root Mean Square (RMS).

RMS is the equivalent mean value of the signal, which is obtained from the squared root of the mean of squared values of $M$ data. Essentially, it measures the signal's overall energy.

$$RMS = \sqrt{\frac{1}{M} \sum_{j=1}^{M} | y_j |^2}$$  (7)

### 3.4.3 Zero Crossings (ZC).

The number of times signal direction changes is measured to obtain

$$ZC = \sum_{j=1}^{M-1} f(y_j)$$

$$here : f(y_j) = \begin{cases} 1 & if \ (y_j > 0 \ and \ y_{j+1} < 0) \\ & or \ (y_j < 0 \ and \ y_{j+1} > 0) \\ 0 & otherwise \end{cases}$$  (8)

### 3.4.4 Slope Sign Change (SSC).

In slope sign change, the slope changes sign for each time it changes. It is an indicator of how many times the signal trend has changed.

$$SSC = \sum_{j=2}^{M-1} f(y_j)$$

$$where : f(y_j) = \begin{cases} 1 & if \ (y_j > y_{j-1} \ and \ y_j > y_{j+1}) \\ & or \ (y_j < y_{j-1} \ and \ y_j < y_{j+1}) \\ 0 & otherwise \end{cases}$$  (9)

### 3.4.5 Variance (VAR).

The measure of the data spread around the mean is called variance. Data points are squared and the average is calculated as the squared difference between the data points and the mean.

$$VAR = \frac{1}{M-1} \sum_{j=1}^{M} y_j^2$$  (10)

### 3.4.6 Difference Absolute Standard Deviation Value (DASDV).

DASDV is the difference between two consecutive windows' absolute standard deviations. The signal's variability over time is measured by this parameter.

$$DASDV = \sqrt{\frac{1}{M-1} \sum_{j=1}^{M-1} (y_{j+1} - y_j)^2}$$  (11)

**3.4.7 Average Amplitude Change (AAC).** Using two consecutive windows, you can calculate the Average Amplitude Change by averaging the absolute changes in amplitude. Over time, the signal changes at a certain rate.

$$AAC = \frac{1}{M} \sum_{j=1}^{M-1} | y_{j+1} - y_j | \tag{12}$$

**3.4.8 Skewness (Skew).** An indicator of the asymmetry of the distribution is the skewness of the data. When skewing is positive, the data is skewed to the right, while when skewing is negative, the data is skewed to the left.

$$Skew = \frac{E[(y - \mu)^3]}{\sigma^3} \tag{13}$$

where, $\mu$, $\sigma$, and $E$ are the mean, standard deviation of the data, and the signal expected value estimator respectively.

**3.4.9 Kurtosis (Kurt).** As a measure of the peakiness of data distributions, kurtosis is used. Generally, the data is flat when the kurtosis is low, and peaked when the kurtosis is high.

$$Kurt = \frac{E[(y - \mu)^4]}{\sigma^4} \tag{14}$$

Four different muscle types are presented here, thus illustrating 44 features across eleven time domain features. In the presence of abnormal knees, the length of the signal differs from that of healthy knees and, consequently, the number of features extracted differs for both classes.

## 3.5 Synthetic Minority Oversampling Technique (SMOTE)

An important aspect of machine learning is using SMOTE to address class imbalance problems in datasets. Chawla et al. [57] proposed this algorithm, which is designed to create synthetic data points in the minority class, which can balance the distribution of the dataset and improve the model's performance. SMOTE is a useful approach when working with unknown datasets because it does not require any prior knowledge of data distribution. The SMOTE algorithm uses a straightforward but effective method to generate these synthetic instances. The first step of the SMOTE algorithm is to identify the k nearest neighbors of each instance in the minority class. The value of k is predetermined and typically set to five [58]. This step helps to identify the most similar instances in the minority class to the instance being analyzed. Once the k nearest neighbors have been identified, SMOTE selects one of these neighbors randomly and generates a synthetic instance by interpolating between the original instance and the chosen neighbor. This interpolation is done by selecting random values for each feature of the original instance and its neighbor. This helps the model to learn the patterns in the minority class and make more accurate predictions. These random values are used to compute a weighted average, which generates a new instance. The weight given to each feature is determined by a random number between 0 and 1. The above steps are repeated until the desired number of synthetic instances have been generated. The number of synthetic instances generated depends on the degree of imbalance in the dataset and the size of the minority class. SMOTE improves the model's ability to learn patterns from the minority class by adding more instances in the minority class, thus balancing the dataset.

## 3.6 Machine learning models

After the completion of oversampling and feature extraction, the extracted features can be used by the classification algorithm for further analysis. The classifiers implemented in this study are Iterative Dichotomiser 3 (ID3), Bootstrap aggregating (Bagging), Random forests (RF), Classification and Regression Trees(CART), Gradient boosting machines(GBC), Support vector machines (SVM), Extra tree (ET), and Multilayer perceptron (MLP).

**3.6.1 Iterative Dichotomiser 3 (ID3).** It is used for the classification of data. Recursively, it partitions the data into smaller and smaller subsets until it reaches the full classification [59]. Since it employs a decision tree of a significant depth, it is more prone to an overfitting problem. It may occur for the high depth of the decision tree.

**3.6.2 Classification and Regression Tree (CART).** It classifies or regresses data based on decision trees [60]. Unlike ID3, CART divides the data based on the Gini index rather than information gain. As a result, priority for separation is first given to the feature having a lesser Gini index:

$$Gini = 1 - \sum_{j=1}^{m} (k_j)^2 \tag{15}$$

where $k_j$ denotes the probability of occurrence of a feature in a particular class.

**3.6.3 Bagging classifier.** It is an ensemble meta-algorithm that combines multiple machine learning algorithms to increase their predictive accuracy [61]. Using bootstrap samples, multiple copies of a learning algorithm are trained on different training data. A final prediction is created by combining the predictions from different learning algorithms.

**3.6.4 Gradient Boosting Classifier (GBC).** It performs prediction by combining multiple decision trees through an ensemble meta-algorithm [62]. It corrects the errors of previous decision trees by sequentially training more decision trees. A greedy tree training algorithm ends when a stopping criterion is met, and the algorithm is terminated based on the stopping criteria.

**3.6.5 Random Forest classifier (RF).** It combines a set of simplistic machine learning models that improves predictive accuracy by combining multiple decision trees [63]. At each node of the decision tree, RF selects features randomly and splits points. It improves the accuracy of the prediction and reduces the variance.

**3.6.6 Support Vector Machines (SVM).** Such algorithms are commonly used in classification and regression tasks [64]. SVMs maximize the margin between two classes by finding the hyperplane between them. The SVM is a powerful algorithm, but it can be computationally intensive.

**3.6.7 Multi-layer Perceptron classifier (MLP).** This machine learning algorithm is used for classification and regression [65]. Multilayer neural networks are comprised of multiple layers of neurons, making them potent and efficient. However, their training poses a challenge due to the non-convex nature of the loss function in MLPs with hidden layers. This non-convexity introduces the presence of multiple local minima, complicating the training process.

**3.6.8 Extra Trees classifier (ET).** It is a machine learning algorithm that improves predictive accuracy through the combination of multiple decision trees [66]. ET produces a final prediction by combining the predictions of each decision tree in a forest. A powerful algorithm, ET, requires a lot of computational power.

## 4 Result and discussions

The evaluation metrics of the classifiers are evaluated without and with the oversampling approach. These approaches are utilized for the problem of imbalanced data in knee abnormality detection. The oversampling approach is used to generate balanced data from the existing original sample of data. The activity signal of the subjects is divided into training and testing sets, with a 10-fold stratified cross-validation technique [67]. It differs from the 10-fold cross-validation. In which the data is randomly partitioned into folds. However, when dealing with imbalanced datasets (such as in knee abnormality detection where certain classes may be underrepresented), it's important to ensure that each fold maintains the same class distribution as the original dataset. Stratified cross-validation achieves this by preserving the percentage of samples for each class in every fold. So, 10-fold stratified cross-validation results in the dataset is divided into 10 equal-sized subsets, and each subset maintains the same class distribution as the original dataset. The model is trained and evaluated 10 times, each time using a different subset as the test set randomly. The number of samples utilized for the classifiers' testing and training set, with and without oversampling, is illustrated in Fig 2. The data for the training of knee abnormal and healthy are 1533, and 342, respectively; the data for the testing are 170, and 38, respectively. Fig 2(B) illustrates the after-oversampling of the data using the SMOTE technique. After the oversample training, the training set data of the healthy knee is increased to equal the abnormal knee (1533), and the testing data remains the same, similar to the without oversample. The obtained results on the oversampled data reflect a significant enhancement in the classifier's performance.

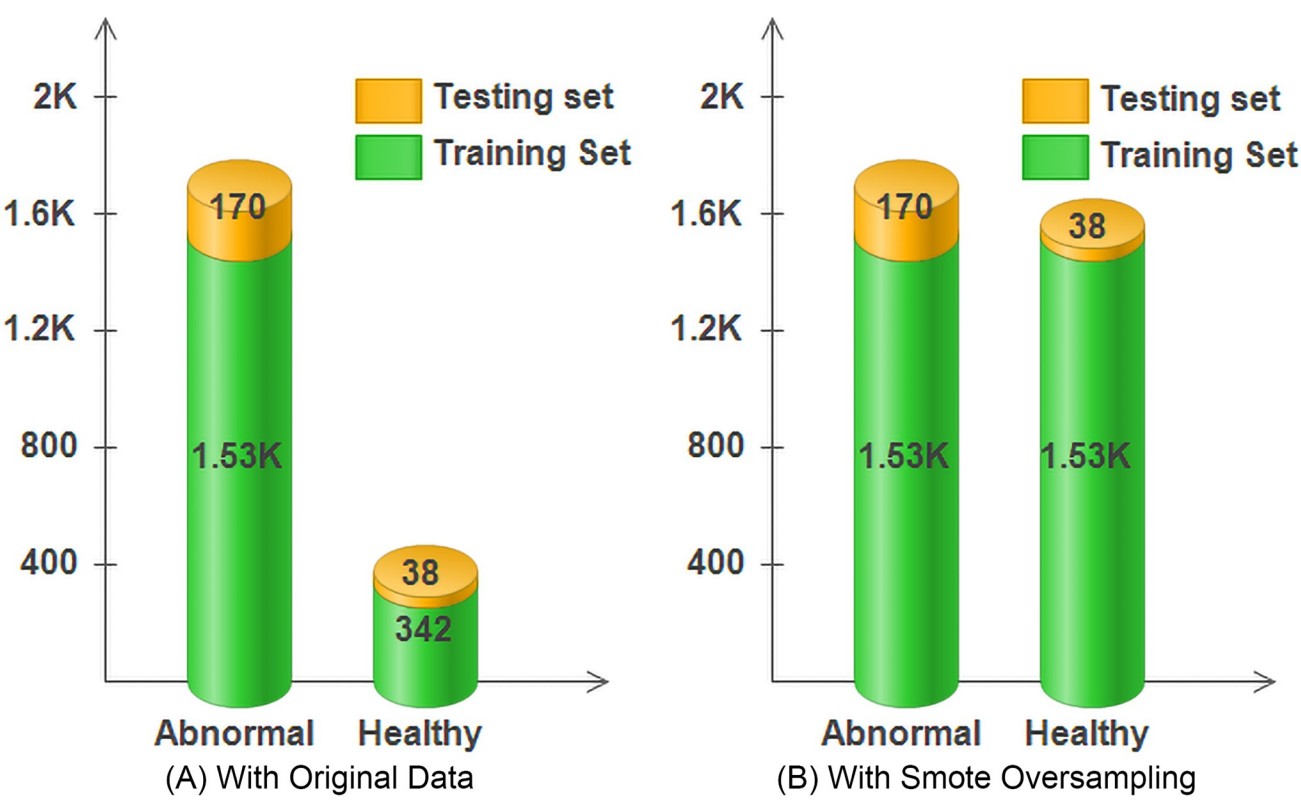

**Fig 2. Samples used to train and evaluate classifiers.** (A) With original data. (B) With SMOTE oversampled data.

In this study, the healthy and knee-abnormal subjects are classified as a binary classification problem. There are four possible outcomes in binary classification. True positives are data points that are correctly predicted to belong to the positive class by a classifier, whereas an accurate classification of a data point into the negative class is known as a true negative (TN). If the classifier makes an incorrect prediction that the data point is positive, it is a false positive (FP) and data points are classified as false negatives (FN) if the classifier predicts them as negatives incorrectly. As a classifier predicts what class the data point will belong to, the terms "true" and "false" indicate whether the prediction matches reality. The two possible classes in which the data point may fall are referred to as "positive" and "negative". These four outcomes can produce a confusion matrix, illustrated in Table 1.

In this study, a total of 8 machine learning models (ID3, CART, GBC, RF, Bagging, SVM, MLP, and ET) are considered for identifying individuals with knee abnormalities. Table 2 presents the details of the parameter settings selected for training the machine learning algorithms. Table 3 provides an exhaustive and detailed comparative analysis of the studied machine learning classifiers. Importantly, these classifiers are evaluated both in their raw form and in conjunction with SMOTE with the studied pre-processing techniques. The assessment metric of accuracy is important, however relying just on accuracy may not be sufficient, especially in situations when there is an imbalance of data [68]. Acknowledging this constraint, the research adopts a broader perspective by going beyond the boundaries of precision to include a range of other measures of performance. One such metric, recall assesses the classifiers' acumen in accurately identifying all pertinent instances within the positive class. Precision scrutinizes the accuracy of the positive predictions generated by these classifiers. In essence, recall and precision serve as critical tools for dissecting and assessing the intricacies of classifier performance. The F1 score harmoniously combines precision and recall into a single measure, effectively striking a balance that duly accounts for both false positives and false negatives. This harmonious synthesis of performance metrics contributes to a more comprehensive and nuanced appraisal of classifier efficacy, particularly in the intricate terrain of imbalanced data distributions.

This study examines the effects of preprocessing techniques and resampling strategies on the performance of classifiers. By conducting a thorough analysis, the study reveals the strengths and weaknesses of each classifier and emphasises the importance of comprehensive performance assessment in the dynamic field of machine learning. Within the confines of this study, the authors have strategically used a combination of preprocessing techniques including Wavelet Denoising (WD), Empirical Mode Decomposition (EMD), and a hybrid fusion that merges both these methodologies. The study revolves around the examination of how these preprocessing techniques wield their influence on classifier performance. Additionally, the study provides valuable insights into the impact of SMOTE on the various performance metrics. It offers a nuanced perspective on how different preprocessing techniques interact with SMOTE, shedding light on the intricacies of model performance.

WD-EEMD excels in extracting richer features from time series data in contrast to traditional methods, effectively capturing both time and frequency domain characteristics. Integrating SMOTE with WD-EEMD ensures the generation of synthetic samples that align with

**Table 1. Confusion matrix.**

|  | Predicted Negative | Predicted Positive |
|---|---|---|
| Actual Negative | True Negatives | False Positives |
| Actual Positive | False Negatives | True Positives |

**Table 2. Parameters of machine learning models.**

| Model | Parameter | Model | Parameter |
|---|---|---|---|
| ID3 | criterion='entropy' | GBC | loss='log_loss' |
| | min_samples_split = 2 | | learning_rate = 0.1 |
| | min_samples_leaf = 1 | | n_estimators = 100 |
| | min_weight_fraction_leaf = 0.0 | | subsample = 1.0 |
| | min_impurity_decrease = 0.0 | | criterion='friedman_mse' |
| | ccp_alpha = 0.0 | | min_samples_split = 2 |
| CART | criterion='gini' | | min_samples_leaf = 1 |
| | min_samples_split = 2 | | min_weight_fraction_leaf = 0.0 |
| | min_samples_leaf = 1 | | max_depth = 3 |
| | min_weight_fraction_leaf = 0.0 | | min_impurity_decrease = 0.0 |
| | min_impurity_decrease = 0.0 | | validation_fraction = 0.1 |
| | ccp_alpha = 0.0 | | tol = 0.0001 |
| Bagging | n_estimators = 10 | | ccp_alpha = 0.0 |
| | max_samples = 1.0 | RF | n_estimators = 100 |
| | max_features = 1.0 | | criterion='gini' |
| ET | n_estimators = 100 | | min_samples_split = 2 |
| | criterion='gini' | | min_samples_leaf = 1 |
| | min_samples_split = 2 | | min_weight_fraction_leaf = 0.0 |
| | min_samples_leaf = 1 | | max_features='sqrt' |
| | min_weight_fraction_leaf = 0.0 | | min_impurity_decrease = 0.0 |
| | max_features='sqrt' | | ccp_alpha = 0.0 |
| | min_impurity_decrease = 0.0 | MLP | hidden_layer_sizes=(100) |
| | verbose = 0 | | solver='adam' |
| | ccp_alpha = 0.0 | | alpha = 0.0001 |
| SVM | C = 1.0 | | learning_rate_init = 0.001 |
| | kernel='rbf' | | power_t = 0.5 |
| | degree = 3 | | max_iter = 200 |
| | gamma='scale' | | tol = 0.0001 |
| | tol = 0.001 | | momentum = 0.9 |
| | cache_size = 200 | | validation_fraction = 0.1 |
| | max_iter=-1 | | beta_1 = 0.9 |
| | decision_function_shape='ovr' | | beta_2 = 0.999 |
| | | | epsilon = 1e-08 |
| | | | n_iter_no_change = 10 |
| | | | max_fun = 15000 |

the local data characteristics, thereby enhancing the diversity and relevance of the samples. This augmentation contributes to improved generalization and heightened classification accuracy, especially in scenarios with imbalanced datasets. However, a notable limitation of this hybrid approach is its potential to substantially escalate computational demands due to the iterative processes involved in both techniques.

Referring to Table 3, in the case of the original dataset (imbalanced without oversampling) employing the wavelet denoising pre-processing technique, the Extra Tree classifier demonstrated an accuracy of 92.0%. In contrast, the ID3, CART, Bagging, Gradient Boosting, Random Forest, SVM, and MLP classifiers showed accuracies of 81.8%, 81.9%, 88.7%, 89.5%, 91.0%, 81.8%, and 84.6%, respectively. Similarly, F-score was 1.5%, 3%, 66.9%, 69.4%, 72.2%,

**Table 3. Classifier in terms of different performance metrics with different pre-processing techniques with SMOTE.**

| Pre-processing | Oversampling Technique | Performance Parameters | ID3 | CART | Bagging | GBC | RF | SVM | MLP | ET |
|---|---|---|---|---|---|---|---|---|---|---|
| WD | Without Oversamplig (Original) | Accuracy | 81.8 | 81.9 | 88.7 | 89.5 | 91.0 | 81.8 | 84.6 | 92.0 |
| | | Recall | 99.9 | 99.8 | 97.1 | 97.4 | 98.5 | 100 | 97.9 | 98.7 |
| | | Precision | 0.8 | 1.6 | 51.3 | 54.2 | 57.4 | 0 | 25.0 | 62.1 |
| | | F-Score | 1.5 | 3 | 66.9 | 69.4 | 72.2 | 0 | 39.0 | 75.9 |
| | With SMOTE Oversapmling | Accuracy | 73.9 | 74.9 | 87.7 | 87.8 | 91.4 | 74.4 | 80.5 | 93.3 |
| | | Recall | 74.7 | 75.8 | 93.0 | 90.0 | 94.5 | 74.3 | 82.0 | 97.0 |
| | | Precision | 70.3 | 71.1 | 63.9 | 77.9 | 77.4 | 74.7 | 73.7 | 76.8 |
| | | F-Score | 71.6 | 72.3 | 75.6 | 83.4 | 85.0 | 74.4 | 77.4 | 85.4 |
| EMD | Without Oversamplig (Original) | Accuracy | 86.2 | 86.4 | 96.0 | 97 | 97.8 | 85.1 | 87.8 | 98.2 |
| | | Recall | 91.4 | 92.8 | 99 | 99.4 | 99.9 | 99.9 | 98.2 | 99.9 |
| | | Precision | 62.6 | 57.6 | 82.6 | 86.1 | 88.7 | 18.7 | 41.1 | 90.5 |
| | | F-Score | 74.2 | 70.6 | 89.9 | 92.1 | 93.8 | 31.1 | 50.6 | 94.9 |
| | With SMOTE Oversapmling | Accuracy | 84.6 | 84.5 | 95.1 | 96.6 | 98.2 | 90.4 | 91.6 | 98.8 |
| | | Recall | 86.7 | 86.8 | 98.1 | 97.5 | 98.9 | 90.0 | 92.3 | 99.5 |
| | | Precision | 75.3 | 73.9 | 81.6 | 92.4 | 95.0 | 91.8 | 88.9 | 95.5 |
| | | F-Score | 79.9 | 79.4 | 89.0 | 94.9 | 96.9 | 90.9 | 90.5 | 97.4 |
| WD-EEMD | Without Oversamplig (Original) | Accuracy | 82.2 | 81.4 | 95.3 | 96.9 | 97.4 | 82.3 | 86.8 | 98.3 |
| | | Recall | 99.8 | 94.0 | 98.9 | 99.4 | 99.7 | 100 | 98.4 | 99.9 |
| | | Precision | 3.4 | 25.3 | 79.2 | 85.8 | 87.1 | 2.9 | 35 | 91.1 |
| | | F-Score | 6.5 | 32.5 | 87.8 | 92.0 | 92.8 | 5.5 | 44.3 | 95.2 |
| | With SMOTE Oversapmling | Accuracy | 80.1 | 81.4 | 94.8 | 96.5 | 98.3 | 89.5 | 90.7 | 99.0 |
| | | Recall | 80.8 | 82.8 | 97.1 | 97.2 | 98.8 | 88.8 | 90.8 | 99.3 |
| | | Precision | 76.8 | 75.3 | 84.7 | 93.4 | 96.1 | 92.9 | 90.3 | 97.6 |
| | | F-Score | 78.5 | 77.9 | 90.4 | 95.2 | 97.4 | 90.7 | 90.4 | 98.4 |

0%, 39.0%, 75.9% for the ID3, CART, Bagging, Gradient Boosting, Random Forest, SVM, MLP and ET classifiers, respectively. In this scenario, the accuracy values appeared satisfactory; however, the F-score values were less compelling. Following the implementation of the SMOTE oversampling technique, it was discovered that the classification accuracy did not show a significant improvement. However, noteworthy enhancement was observed in the F-Score values compared to those obtained from the original dataset. When employing SMOTE oversampling, the accuracies for the ID3, CART, Bagging, Gradient Boosting, Random Forest, SVM, MLP, and ET classifiers were 73.9%, 74.9%, 87.7%, 87.8%, 91.4%, 74.4%, 80.5%, and 93.3%, respectively. Correspondingly, the F-scores were 71.6%, 72.3%, 75.6%, 83.4%, 85.0%, 74.4%, 77.4%, and 85.4% for the mentioned classifiers. Likewise, the F-score values for the examined machine learning classifiers exhibited higher values when employing the SMOTE oversampling technique along with the EMD and hybrid WD-EEMD pre-processing techniques compared to the original dataset (imbalanced dataset). These findings emphasize the multifaceted nature of preprocessing techniques and their intricate interplay with resampling methods like SMOTE. They underscore the importance of considering a holistic approach when evaluating machine learning models, taking into account not only the choice of classifier but also the preprocessing steps to attain the most accurate and robust results, especially when tackling data with imbalances and complexities.

The WD-EEMD-based hybrid approach is of particular interest. It combines the strengths of both the pre-processing techniques and when integrated with the ET classifier, gives an accuracy of 98.3% and the F-score value is also 95.2% without SMOTE. After including the

SMOTE, the accuracy rises to 99.0% and F-score improves to 98.4% showcasing the advantage of the proposed technique. These findings underscore the versatility and potency of various preprocessing techniques, each with its unique advantages, and highlight the potential for even greater performance improvements through hybridization. It reaffirms that the careful selection and combination of preprocessing methods can significantly elevate the accuracy and robustness of machine learning models, particularly when dealing with complex and imbalanced datasets The research findings strongly indicate that the synergistic application of preprocessing techniques, notably the hybrid WD-EMD method, yields substantial enhancements in classifier accuracy. It is noteworthy that the Extra Tree classifier consistently emerges as the top performer throughout these experiments. Therefore, the suggested S-WD-EEMD with the Extra Tree classifier can be employed to detect abnormalities in the knee when there is an imbalance in the EMG signals of lower limb muscles.

Fig 3 displays the confusion matrices of the ET classifier with different pre-processing techniques for original data as well as oversampled data. A confusion matrix offers a structured overview of a classification method's performance, detailing the correspondence between actual and predicted labels generated by the model. Fig 3(A) shows the confusion matrix of the Extra Tree classifier with the original data preprocessed by wavelet denoising. It can be seen that 98.9% samples of Abnormal individuals and 60.5% samples of healthy individuals are

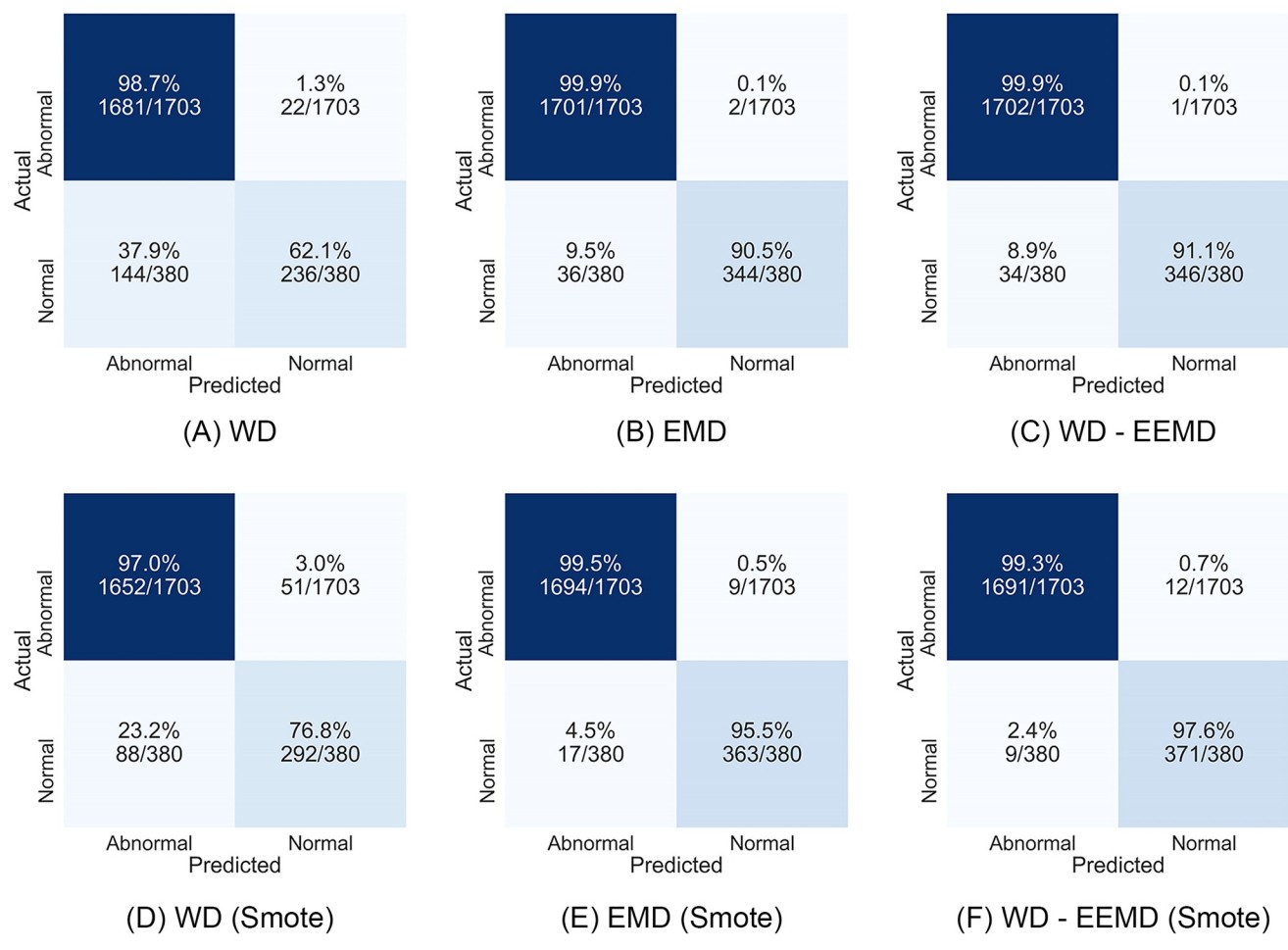

**Fig 3. Confusion matrix with Extra Trees classifier.** (A-C) With original data. (D-F) With SMOTE oversampled data.

correctly predicted while 39.5% samples of healthy individuals are predicted as abnormal and 1.1% samples of abnormal individuals are predicted as healthy. Similarly, Fig 3(B) and 3(C) also presents the confusion matrix of ET classifiers with original data with EMD and WD-EEMD preprocessing technique, and Fig 3(D)–3(F) present the confusion matrix of ET classifier with oversampled data with WD, EMD, and WD-EEMD preprocessing technique. Based on the examination of the confusion matrices, it can be deduced that the utilized classification models tended to favor the predominant class (abnormal individuals) during training on the original dataset. Nevertheless, when augmenting the data of the minority class using oversampling approaches, the classification models ceased to exhibit a bias towards a singular class.

To support the findings presented in Table 3 and provide additional insights, a Receiver Operating Characteristic (ROC) analysis was conducted and illustrated in Fig 4. ROC analysis serves to validate the efficacy of a statistical machine learning model for a given problem by plotting a curve representing the true positive rate against the false positive rate. Each point on the ROC curve corresponds to an instance of a confusion matrix. Ideally, a model is considered 100% accurate if its ROC curve exhibits a point in the top-left corner of the plot. Observing

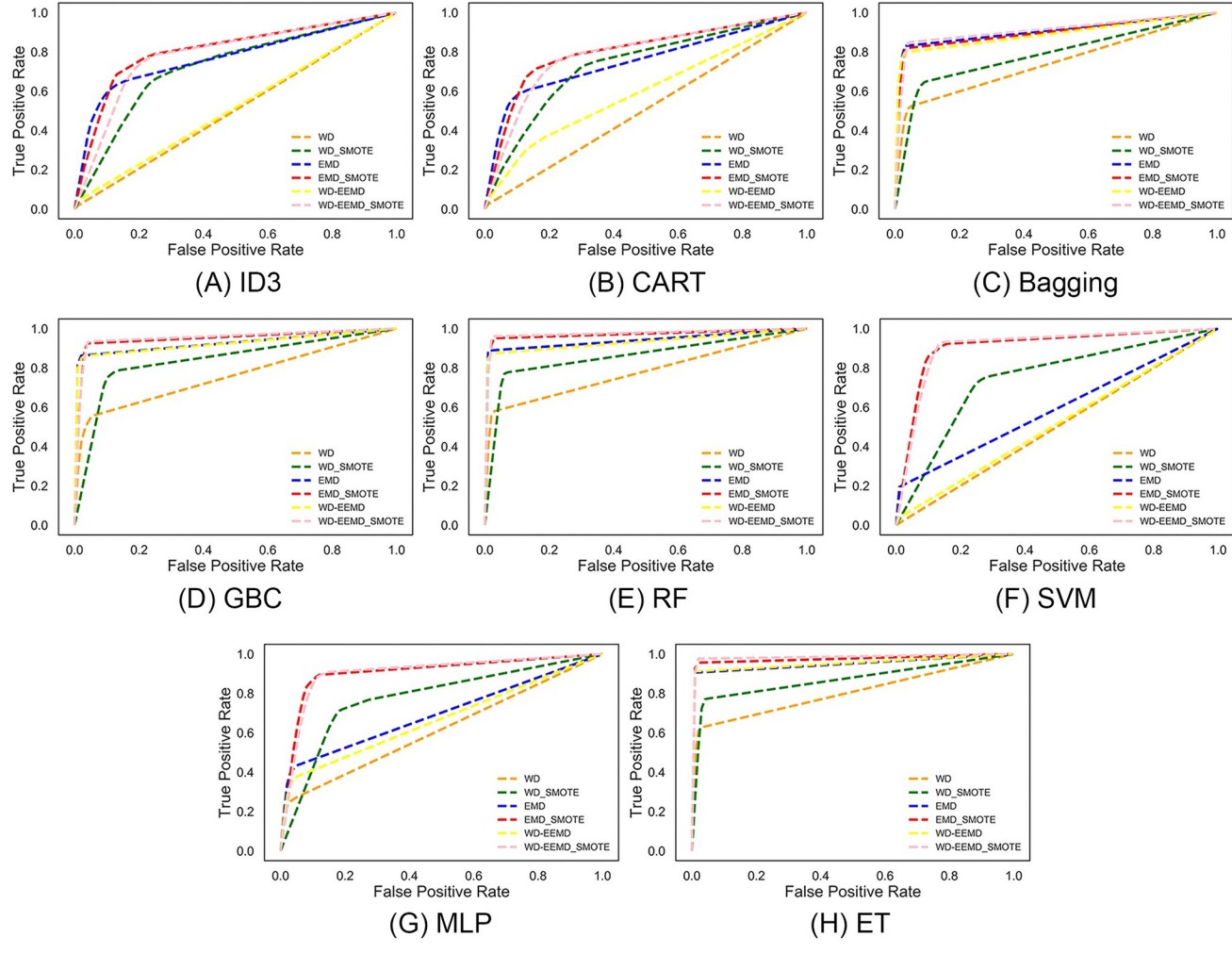

**Fig 4. ROC curve.**

**Table 4. Assessing the effectiveness of the proposed methodology by comparing its performance with existing literature studies utilizing similar datasets (in %).**

| Approach | SMOTE + ET [43] | ADAYSN + ET [43] | SVM SMOTE + ET [43] | XGBoost [69] | Proposed Method |
|---|---|---|---|---|---|
| Accuracy | 93.1 | 92.2 | 93.2 | 96 | **99.0** |
| F-Score | 85.3 | 84.2 | 85.1 | 94 | **98.4** |

Fig 4, it is apparent that before applying SMOTE, when the data exhibits a non-uniform distribution, the ROC curves for all models slightly shift towards the false positive region. This suggests a bias towards a specific class due to data imbalance, a common challenge in learning with imbalanced data. However, upon applying SMOTE, there is a noticeable shift towards the ideal point (0,1) on the ROC plot, indicating improved model performance. Furthermore, the simultaneous application of WD and EEMD pre-processing with SMOTE leads to further improvement in model performance, as evidenced by the ROC curve shifting closer to the ideal point. This underscores the efficacy of these pre-processing techniques in enhancing model identification across all testing data instances. These findings emphasize the critical importance of addressing data imbalance to achieve satisfactory classification algorithm performance. Moreover, the utilization of WD-EEMD pre-processing techniques not only mitigates outliers but also brings numerical features within a considerable range, thereby facilitating enhanced classification.

Very few studies have been published regarding the detection of knee abnormalities in the literature using the same dataset. In our previous research, we applied various oversampling approaches such as SMOTE, ADAYSN, and SVM SMOTE, and evaluated their performance using the Extra Tree machine learning classifier as presented in Table 4. Rani et al. [69] also utilized the same dataset to classify subjects with knee abnormalities using machine learning approaches. Their XGBoost classifier achieved an accuracy of 96% with an F-Score of 94%. Table 4 compares our suggested model's performance with previous studies on the same dataset. Our findings suggest that the proposed technique performs better in identifying individuals with knee abnormalities.

## 5 Conclusion and future scope

In the context of knee problem detection utilizing surface electromyography (sEMG) signals, our research tackles a significant obstacle in the field of medical data classification, which is the existence of unbalanced. We have found that the intrinsic signal noise, in addition to the substantial heterogeneity in sEMG signal durations that exists between healthy persons and those with knee problems, makes it even more difficult to make an accurate diagnosis. To address this difficulty, a hybrid technique S-WD-EEMD is proposed in this study. This approach combines the Wavelet Decomposition-enhanced Ensemble Empirical Mode Decomposition (WD-EEMD) with the Synthetic Minority Oversampling Technique (SMOTE).

According to the findings of our study, the S-WD-EEMD methodology has several significant benefits over the alternatives that are currently being used. SMOTE allows us to successfully address the imbalance problem that is inherent in the dataset, which in turn enables us to classify knee anomalies more reliably and accurately. Second, the combination of WD and EEMD optimizes the process of extracting features from sEMG data. This allows for the capture of significant signal properties while simultaneously reducing the impact of noise. Our experimental findings demonstrate that the performance of the classifier is greatly enhanced by the synergistic combination of oversampling and preprocessing procedures.

The results demonstrate a significant improvement in classification accuracy, sensitivity, and specificity when compared to the methodologies that have been traditionally used. Furthermore, the S-WD-EEMD technique demonstrates excellent performance in the management of unbalanced data settings, surpassing models that are considered to be state-of-the-art in diagnostic tasks involving knee abnormalities. In medical settings, where the prompt and accurate diagnosis of knee problems is of the utmost importance, this breakthrough offers a great deal of potential for enhancing patient care and diagnostic accuracy. In summary, the proposed model can be used to identify individuals with potential knee injuries (anterior cruciate ligament and meniscus damage), or possibly those predisposed to knee injuries depending on how advanced the change in neuromuscular EMG functioning is relative to the tissue structural damage. These individuals could then be referred for clinical examination and either prevention or rehabilitation, depending on the level of injury.

In this study, a single offline sEMG dataset was utilized to evaluate the suggested approach. To further establish the clinical applicability of our method, future research endeavors may entail the validation of this method using a larger dataset. Such larger data would provide invaluable insights into the applicability and efficacy of our approach in clinical settings, thereby enhancing its clinical utility. Furthermore, the proposed hybrid approach combining Wavelet Decomposition (WD), Ensemble Empirical Mode Decomposition (EEMD), and Synthetic Minority Over-sampling Technique (SMOTE) can potentially be extended to other medical applications characterized by imbalanced data. This suggests its potential for addressing similar challenges in diverse medical contexts, thereby expanding its applicability and relevance beyond the scope of this study.

## Author Contributions

**Conceptualization:** Ankit Vijayvargiya, Rajesh Kumar.

**Data curation:** Ankit Vijayvargiya.

**Formal analysis:** Ankit Vijayvargiya, Aparna Sinha, Ashutosh Jena.

**Software:** Ankit Vijayvargiya, Ashutosh Jena.

**Supervision:** Rajesh Kumar, Kieran Moran.

**Validation:** Naveen Gehlot, Rajesh Kumar, Kieran Moran.

**Visualization:** Ankit Vijayvargiya, Aparna Sinha.

**Writing – original draft:** Ankit Vijayvargiya, Aparna Sinha, Naveen Gehlot.

**Writing – review & editing:** Rajesh Kumar, Kieran Moran.

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
