## [Decision Letter · Decision Letter 0]

22 Feb 2024

PONE-D-23-43284S-WD-EEMD: A Hybrid Framework for Imbalanced sEMG Signal Analysis in Diagnosis of Human Knee AbnormalityPLOS ONE

Dear Dr. Vijayvargiya,

Thank you for submitting your manuscript to PLOS ONE. After careful consideration, we feel that it has merit but does not fully meet PLOS ONE’s publication criteria as it currently stands. Therefore, we invite you to submit a revised version of the manuscript that addresses the points raised during the review process.

**ACADEMIC EDITOR: Major Revision **

We look forward to receiving your revised manuscript.

Kind regards,

Shahid Akbar, PhD

Academic Editor

PLOS ONE

Journal Requirements:

4. PLOS requires an ORCID iD for the corresponding author in Editorial Manager on papers submitted after December 6th, 2016. Please ensure that you have an ORCID iD and that it is validated in Editorial Manager. To do this, go to ‘Update my Information’ (in the upper left-hand corner of the main menu), and click on the Fetch/Validate link next to the ORCID field. This will take you to the ORCID site and allow you to create a new iD or authenticate a pre-existing iD in Editorial Manager. Please see the following video for instructions on linking an ORCID iD to your Editorial Manager account: https://www.youtube.com/watch?v=_xcclfuvtxQ.

Reviewers' comments:

Reviewer's Responses to Questions

**Comments to the Author**

1. Is the manuscript technically sound, and do the data support the conclusions?

Reviewer #1: Yes

Reviewer #2: Yes

2. Has the statistical analysis been performed appropriately and rigorously? 

Reviewer #1: Yes

Reviewer #2: Yes

3. Have the authors made all data underlying the findings in their manuscript fully available?

Reviewer #1: Yes

Reviewer #2: Yes

4. Is the manuscript presented in an intelligible fashion and written in standard English?

Reviewer #1: Yes

Reviewer #2: Yes

5. Review Comments to the Author

Reviewer #1: In order to improve the quality of paper, the authors are advised to incorporate the following recommendations.

1) the authors are claiming a very high performance accuracy of 99%, how the authors handle overfitting issue of the model to validate the generalization of the model.

2) to prove the effectiveness of the model, the thorough comparison of the proposed model with existing studies are needs to be performed.

3) the discussion is not clear, authors are advised to provide the advantages and limitations of the proposed model.

4) for the reader concerns, the recent predictors should be cited such as iAFP-gap-SMOTE, and AIPs-SnTCN in SMOTE section. additionally iAFPs-EnC-GA model should be cited and discuss in the wavelet transformation section.

5) the conclusion needs more improvements in term of contribution, advantages, achieved results than existing model.

Reviewer #2: 1. The feature extraction methods are very shortly explained, I suggest to add more details to the readers.

2. The optimal parameters selected for training the machine learning models should be provided in the form of a table.

3. What should be the real life applications developed based on the proposed model, authors should discuss.

4. How many features were extracted after applying proposed methods, authors should discuss in the paper.

5. In order to show the effectiveness of the proposed model, the ROC analysis of the training models is highly recommended.

6. The recent computational models such as 10.1109/TNSRE.2016.2636367, 10.3389/fgene.2020.539227, 10.1109/ACCESS.2021.3062291

6. PLOS authors have the option to publish the peer review history of their article (what does this mean?). If published, this will include your full peer review and any attached files.

Reviewer #1: No

Reviewer #2: No

---

## [Author Response · Author response to Decision Letter 0]

12 Mar 2024

The authors would like to extend their sincere appreciation to the reviewers and editor for generously dedicating their time and offering invaluable recommendations to enhance the manuscript submitted to the “PLOS ONE” journal. Every effort has been made by the authors to integrate all suggestions provided by the reviewers and address all inquiries raised by them. For a detailed response to the comments, the reply letter is attached.

---

## [Decision Letter · Decision Letter 1]

14 Mar 2024

S-WD-EEMD: A Hybrid Framework for Imbalanced sEMG Signal Analysis in Diagnosis of Human Knee Abnormality

PONE-D-23-43284R1

Dear Dr. Vijayvargiya,

We’re pleased to inform you that your manuscript has been judged scientifically suitable for publication and will be formally accepted for publication once it meets all outstanding technical requirements.

Kind regards,

Shahid Akbar, PhD

Academic Editor

PLOS ONE

Additional Editor Comments (optional):

Reviewers' comments:

Reviewer's Responses to Questions

**Comments to the Author**

1. If the authors have adequately addressed your comments raised in a previous round of review and you feel that this manuscript is now acceptable for publication, you may indicate that here to bypass the “Comments to the Author” section, enter your conflict of interest statement in the “Confidential to Editor” section, and submit your "Accept" recommendation.

Reviewer #1: All comments have been addressed

Reviewer #2: All comments have been addressed

2. Is the manuscript technically sound, and do the data support the conclusions?

Reviewer #1: Yes

Reviewer #2: Yes

3. Has the statistical analysis been performed appropriately and rigorously? 

Reviewer #1: Yes

Reviewer #2: Yes

4. Have the authors made all data underlying the findings in their manuscript fully available?

Reviewer #1: Yes

Reviewer #2: Yes

5. Is the manuscript presented in an intelligible fashion and written in standard English?

Reviewer #1: Yes

Reviewer #2: Yes

6. Review Comments to the Author

Reviewer #1: the authors have successfully all of my concerns and i have no further comments... my decision is to accept

Reviewer #2: all my previous concerns have been addressed and i have no further comments. hence i recommend to publish this article from my side

7. PLOS authors have the option to publish the peer review history of their article (what does this mean?). If published, this will include your full peer review and any attached files.

Reviewer #1: No

Reviewer #2: No

---

## [Editor Report · Acceptance letter]

8 May 2024

PONE-D-23-43284R1 

PLOS ONE

Dear Dr. Vijayvargiya, 

I'm pleased to inform you that your manuscript has been deemed suitable for publication in PLOS ONE. Congratulations! Your manuscript is now being handed over to our production team.

Kind regards, 

on behalf of

Dr. Shahid Akbar 

Academic Editor

PLOS ONE